# Beyond First-Line Immunotherapy: Potential Therapeutic Strategies Based on Different Pattern Progressions: Oligo and Systemic Progression

**DOI:** 10.3390/cancers13061300

**Published:** 2021-03-15

**Authors:** Arsela Prelaj, Chiara Carlotta Pircher, Giacomo Massa, Valentino Martelli, Giulia Corrao, Giuseppe Lo Russo, Claudia Proto, Roberto Ferrara, Giulia Galli, Alessandro De Toma, Carlo Genova, Barbara Alicja Jereczek-Fossa, Filippo de Braud, Marina Chiara Garassino, Sara Elena Rebuzzi

**Affiliations:** 1Medical Oncology Department, Fondazione IRCCS Istituto Nazionale Tumori, via Giacomo Venezian 1, 20133 Milan, Italy; chiara.pircher@istitutotumori.mi.it (C.C.P.); giacomo.massa@istitutotumori.mi.it (G.M.); Giuseppe.LoRusso@istitutotumori.mi.it (G.L.R.); Claudia.proto@istitutotumori.mi.it (C.P.); roberto.ferrara@istitutotumori.mi.it (R.F.); giulia.galli@isitutotumori.mi.it (G.G.); alessandro.detoma@istitutotumori.mi.it (A.D.T.); filippo.debraud@istitutotumori.mi.it (F.d.B.); marina.garassino@istitutotumori.mi.it (M.C.G.); 2Department of Electronics, Information, and Bioengineering, Polytechnic University of Milan, Piazza Leonardo Da Vinci 32, 20133 Milan, Italy; 3Oncologia Medica 1, IRCCS Ospedale Policlinico San Martino, Largo Rosanna Benzi 10, 16132 Genova, Italy; valentino.martelli@hsanmartino.it (V.M.); or saraelena89@hotmail.it (S.E.R.); 4Division of Radiation Oncology, IEO, European Institute of Oncology IRCCS, via Ripamonti 435, 20141 Milan, Italy; giulia.corrao@ieo.it (G.C.); barbara.jereczek@ieo.it (B.A.J.-F.); 5Department of Oncology and Hemato-Oncology, University of Milan, via Festa del Perdono, 7, 20122 Milan, Italy; 6UO Clinica di Oncologia Medica, IRCCS Ospedale Policlinico San Martino, Largo Rosanna Benzi 10, 16132 Genova, Italy; carlo.genova@hsanmartino.it; 7Dipartimento di Medicina Interna e Specialità Mediche (DiMI), Università degli Studi di Genova, Viale Benedetto XV 6, 16132 Genoa, Italy

**Keywords:** non-small cell lung cancer, oligoprogression, immune checkpoint inhibitor, immunotherapy, local therapy, resistance

## Abstract

**Simple Summary:**

The aim of this innovative review is to highlight the treatment strategies beyond immune-checkpoint inhibitor (ICI)-based first-line therapy failure according to different patterns of progression (i.e., oligo or systemic progression) and to discuss the ongoing and potential future therapeutic approaches to overcome resistance to immunotherapy. Many therapeutic strategies can be adapted in advanced non-small cell lung cancer patients with oligo and systemic progression to personalize the treatment approach based on to re-characterization of the tumors, previous ICI response and type of progression.

**Abstract:**

First-line immune-checkpoint inhibitor (ICI)-based therapy has deeply changed the treatment landscape and prognosis in advanced non-small cell lung cancer (aNSCLC) patients with no targetable alterations. Nonetheless, a percentage of patients progressed on ICI as monotherapy or combinations. Open questions remain on patients’ selection, the identification of biomarkers of primary resistance to immunotherapy and the treatment strategies to overcome secondary resistance to first-line immunotherapy. Local ablative approaches are the main therapeutic strategies in oligoprogressive disease, and their role is emerging in patients treated with immunotherapy. Many therapeutic strategies can be adapted in aNSCLC patients with systemic progression to personalize the treatment approach according to re-characterization of the tumors, previous ICI response, and type of progression. This review’s aim is to highlight and discuss the current and potential therapeutic approaches beyond first-line ICI-based therapy in aNSCLC patients based on the pattern of disease progression (oligoprogression versus systemic progression).

## 1. Introduction

In recent years, immune checkpoint inhibitors (ICIs) have dramatically changed the treatment landscape of advanced non-small cell lung cancer (aNSCLC). Most aNSCLC patients do not harbor a targetable alteration (non-oncogene aNSCLC); thus, immunotherapy, as single agent or in combination with other drugs (immuno-chemotherapy (CT)—ICI-CT—or immuno-immunotherapy—ICI-ICI) are the mainstay treatment based on the programmed death-ligand 1 (PD-L1)’s expression [1,2].

Despite the survival advantage, only a percentage of aNSCLC patients respond to single-agent ICI (20–30%) and ICI-based combinations (50–60%), while most patients experience disease progression [1,2]. The treatment choice after failure of first-line ICI-based therapy depends on previous treatment, type of response and progression, the burden of disease, and patient performance. Several unanswered questions include patients’ selection and the identification of prognostic and predictive biomarkers of primary and secondary resistance to ICIs. The establishment of therapeutic strategies to overcome failure of ICIsand to extend its benefit in non-responding and progressing patients is a clinical unmet need and a critical area of research [3,4].

This review aimed to highlight the treatment strategies beyond failure ICI-based first-line therapy according to different patterns of progression (i.e., oligo or systemic progression) and to discuss the ongoing and potential future therapeutic approaches to overcome resistance to immunotherapy.

## 2. Definition and Biology of Oligoprogression in aNSCLC

The term “oligoprogression” refers to the progression of a limited number of metastatic lesions in the context of a well-controlled metastatic disease [2,5]. This concept is related to the wider definition of “oligometastatic” disease (“oligo” = “few”) that refers to a limited number of metastases and/or metastatic sites characterized by a more indolent behavior than a polymetastatic disease [5,6]. Although there is no consensus on the appropriate cut off to define the oligometastatic state, generally up to 3–5 lesions in 1–3 organs are defined in international guidelines and are commonly accepted [7,8,9].

The most available evidence for oligoprogression is reported in aNSCLC patients treated with tyrosine kinase inhibitors (TKIs). Their inter- and intra-tumors heterogeneity are the biological basis of this clinical scenario.

Tyrosine kinase inhibitors selectively favor tumor clones that are intrinsically resistant (primary resistance) or induce changes in tumors’ phenotype (secondary resistance) [5]. Few studies are available regarding oligoprogression on ICI-treated patients. In this case, oligoprogression might represent local immune tolerance, even though mechanisms of resistance are more heterogeneous and are not well-defined [10].

The incidence of oligoprogression is 15–47% with TKI and 10–20% with ICI or ICI-based combinations [11,12,13]. The most common oligometastatic and oligoprogressive sites include brain, lung, lymph nodes, liver, and adrenal glands [13,14].

## 3. Local Ablative Therapies in Oligoprogressive aNSCLC

Many studies have been conducted on the efficacy and safety of local ablative therapies (LAT), including radiotherapy (RT), surgery, and radiofrequency ablation (RFA), in oligometastatic and oligoprogressive aNSCLC patients [5,10,15].

In the oligometastatic disease, such approaches could be potentially curative, while in oligoprogressive patients they allow to eliminate foci of resistance, continue systemic treatment and postpone further therapies increasing disease control and overall survival (OS) [5,10,15].

Radiotherapy, especially stereotactic body radiation therapy (SBRT), is the most used and studied LAT for cranial and extra-cranial metastases, as it is non-invasive, well-tolerated, and permits fewer interruptions of systemic therapy [5,10,15].

In ICI-treated patients, RT is the most promising LAT in oligoprogressive patients for its synergistic effect on enhancing the anti-tumor action of the immune system and overcoming ICIs’ resistance [16,17]. These pathogenetic mechanisms are the basis of the abscopal effect, a rare event consisting of an immune-mediated response to RT of metastases distant from the irradiated site (from “ab scopus”, i.e., “away from the target”) [16]. Multiple questions remain regarding optimal patient selection, choice of RT schedule, and sequencing between RT and systemic therapy [16].

The second most important LAT is surgery (metastasectomy). However, its efficacy was shown to improve with the combination of other techniques, especially RT [10,18]. The strongest evidence in LAT surgery refers to brain metastases surgery, while the evidence lacks in extracranial metastases, which include mainly lung and adrenal gland ones [10,18]. Other LATs, with less evidence, including RFA and cryoablation, especially on lung and liver metastases [10,18].

## 4. Local Ablative Therapies in Oligoprogressive aNSCLC Treated with ICIs

### 4.1. Current Evidence

Local ablative therapies represent a consolidated approach in oligoprogressive oncogene-addicted aNSCLC, but its role in immunotherapy has been poorly addressed. Although this therapeutic strategy is not a standard approach with ICIs, its use in clinical practice is an extrapolation of data from patients treated with CT or TKI.

Few case reports on oligoprogression of ICI-treated aNSCLC patients showed that LAT (RT and/or surgery) continuing immunotherapy beyond progression was associated with successful disease control and long-term survival benefit [19,20,21]. The clinical evidence on LAT in oligoprogressed ICI-treated aNSCLC patients comes mainly from retrospective analysis (Table 1) [13,22,23,24,25,26].

In 2016, Bledsoe et al. [22] reported that RT for oligoprogressive lesions was safe and offered good local control at months 6 and 12 in 92% and 85% of 21 aNSCLC patients receiving ICIs.

In a retrospective analysis on 81 ICI-treated patients, oligoprogression was observed in approximately 10% of progressed patients, and 50% of them received LAT with durable clinical benefit [23].

A higher number of oligoprogression (88%) was reported by Gettinger et al. [24] among 26 aNSCLC patients progressing on ICIs. A similar percentage of patients (58%) received LAT (mainly RT) to oligoprogressive site(s), and 73% of patients continued immunotherapy with prolonged benefit. Similarly, another analysis, on 27 aNSCLC patients treated with LAT (RT and surgery) for oligoprogression on anti-PD-1 agents, reported that ICI was continued beyond progression in 22 patients (81%) and mPFS (median Progression-Free Survival) after LAT was 13.2 months [25].

In a larger retrospective study (100 patients progressing on pembrolizumab), 18 (18%) patients were treated beyond progression and among them, 13 patients (72%) had oligoprogression and nine (69%) patients were treated with the addition of RT [26]. The combination of pembrolizumab beyond progression and RT was associated with high post-progression survival.

Recently, Rheinheimer et al. [13] conducted a retrospective analysis on 372 aNSCLC patients treated with ICI-based therapy (ICI alone or in combinations with other drugs). About 13% of patients developed oligoprogression and LATs were more frequently offered to patients with brain metastases than extracranial cases (72% versus 49%). Moreover, LATs were more frequently used in patients treated with ICI–CT compare to ICI monotherapy (90% versus 50%).

Finally, a retrospective international multicenters register study (TOaSTT) collected data on NSCLC patients who received SBRT and systemic therapy based on ICI/TKI. A total of 192 lesions on 108 patients were analyzed. Of the latter, 60% received TKI, while 31% received ICI and 8% bevacizumab (antiangiogenic drug). Oligoprogressive and oligopersistent (patients with an existing oligometastasis at baseline) patients showed significant improvement in OS. The PFS was superior in the oligoprogressive group (20.1 versus 7 versus 4.4 months, respectively) [27].

All these studies showed that for oligoprogressive aNSCLC patients treated with immunotherapy, LAT associated with continuation immunotherapy beyond PD appeared to be a safe therapeutic option providing promising long-term survival. However, despite the encouraging data, most of these studies are small retrospective analyses and larger prospective studies are needed.

### 4.2. Ongoing Trials

In addition to the abovementioned analyses (Section 4.1), there are several ongoing prospective clinical trials assessing the addition of RT to oligoprogressive ICI-treated aNSCLC patients (Table 2).

Two studies are evaluating the efficacy and safety of the combination of a PD-L1 inhibitor (avelumab or atezolizumab) and RT after progression on a PD-1 inhibitor (nivolumab or pembrolizumab) (NCT03158883, NCT04549428).

The UCDCC#270 is an early phase I single-center study which the aim of assessing the combination of avelumab and SBRT in 26 participants divided into “non-responders” and “progressors” to immunotherapy who previously failed platinum-based CT (NCT03158883).

A similar study is the NCT04549428 trial, a multicenter phase II, single-arm study evaluating the combination of atezolizumab with palliative RT (8 Gy single-fraction) in aNSCLC patients who oligoprogressed (≤4 lesions) upon monoimmunotherapy received in sequence after CT or in combination with CT.

In both studies, the primary endpoint is the objective response rate (ORR), while OS and progression-free survival (PFS) are secondary endpoints.

Five other ongoing trials are designed to evaluate the efficacy and safety of the RT and ICI combination in patients with oligoprogressive disease continuing the same ICI beyond PD.

The SUPPRESS-NSCLC study is a phase II trial which will randomize 68 aNSCLC patients who oligoprogressed (≤5 lesions) on ICI or TKI (at any line) to receive SBRT while continuing the current systemic therapy versus standard of care (begin next-line systemic therapy, best supportive care, continue current systemic line) (NCT04405401).

Another randomized trial is the phase II NCT04485026 study designed to evaluate the efficacy of local consolidative RT versus second-line therapy in aNSCLC patients who oligoprogressed (≤4 lesions) on ICI-based first-line therapy after having experienced response or stable disease.

The NCT03693014 is an additional phase II monocentric trial assessing the efficacy of hypofractionated RT in oligoprogressive ICI-treated patients. Unlike other studies, this trial will include different advanced tumors types, including aNSCLC, treated with different ICIs. The enrolled patients must have evidence of limited progression (≤5 lesions) and will receive SBRT to a maximum of three lesions, while continuing immunotherapy, the hypofractionated RT seems to be more immunogenic.

The phase II NCT03406468 trial will assess the efficacy of RT to a single lesion in 40 aNSCLC patients who progressed on ICI monotherapy or ICI–CT combination and have previously achieved stable disease or response to immunotherapy.

Finally, the NCT04492969 trial will be the largest study (320 estimated participants) with the aim to prospectively evaluate the pattern progression in aNSCLC patients after response to ICI. Moreover, the feasibility and clinical value of RT in oligoprogressive lesions (1–3 lesions in 1–2 organs) after ICI will be investigated.

Oligoprogression in aNSCLC patients treated with immunotherapy is an uncommon finding in clinical practice; however, researchers show an increasing interest in this setting, since it can be associated with a good prognosis if treated properly. This is reflected in the growing number of ongoing or planned clinical trials even though large multicenter, randomized, phase III clinical trials are still needed.

## 5. Systemic Progression

### 5.1. Timing of Systemic Progression

In this review, systemic progression on ICI–CT was divided into three categories based on the timing of progression: early systemic progression (ES-PD), characterized by the lack of response to immunotherapy and a disease progression occurring within the first 3 months of treatment initiation; intermediate systemic progression (IS-PD) defined as a disease progression occurring between 3 months and 2 years from the start of treatment; late systemic progression (LS-PD) defined as a disease progression occurring at least after 2 years of ICI treatment.

The biological mechanisms underlying resistance to immunotherapy are not well defined and the complexity of the tumors microenvironment (TME) and the immune system is translated into different types and mechanisms of ES-PD [28].

### 5.2. Early Progression Mechanisms and Definitions

The immune escape mechanisms underlying primary resistance to immunotherapy are present at baseline immunotherapy and regard the defective “ignition” (priming defective mechanism) or the development and consolidation (adaptive immune resistance) of the immune response [29,30] (Table 3).

On the contrary, the acquired resistance to immunotherapy develops after the acquisition of new tumors escape mechanisms during treatment after an initial phase of response or stable disease (secondary resistance) [24,28].

These mechanisms concern the decreased production and expression of tumors antigens (e.g., human leukocyte antigen (HLA) class I defects [31,32]), epigenetic modifications [33,34], genetic mutations (e.g., mutations of MAPK pathway [35], loss of PTEN [36] with an increase of PI3K [37,38], expression of WNT/β-catenin [39], altered IFN-α pathway [40], EGFR mutations [41], MYC overexpression [42]), alterations of PDJ amplicon on chromosome 9, which codes for PD-L1/2 and JAK2, and alterations of a gene set called IPRES [43], related to a mesenchymal transformation [28]. The intrinsic mechanisms inclued also TME alteration, including molecules and cells of the tumor stroma, immune-regulatory cell (Tregs [44], MDSCs [45], M2 macrophages [46]), immune checkpoints. And soluble molecules, such as IFN-α, which leads to the production of IDO [47] and CEACAM-1 [28,48].

Clinical studies are lacking in describing resistance mechanisms ICI–CT combination and also data indicating for treatment beyond progression on first-line ICI–CT is currently limited [49,50,51,52,53].

### 5.3. Hyperprogression and Fast Progression

Hyperprogressive disease (HPD) is described as the acceleration of the disease progression during ICI compared to the natural history of the tumors, associated with a rapid worsening of clinical conditions within the first imaging evaluation and poor prognosis [54]. Specific criteria were identified by Lo Russo et al. [55] and are used in clinical practice for the definition of HPD. Despite it, the biological mechanisms are still unknown, HPD has become an emerging clinical issue in the immunotherapy era [54]. Its prevalence is 10–20% in ICI-treated aNSCLC and most literature evidence regard single-agent ICI, while few data are available with the ICI-based combinations and its prevalence [55,56]. To date, HPD did not seem to correlate with a particular patient, tumor or treatment characteristic, and no predictive markers are available [57,58].

A different clinical entity described is fast progression (FP), which is a progression within the first radiologic evaluation or no later than 12 weeks but not classifiable as HPD criteria [59]. According to some small evidence on the biological mechanisms, HPD is associated with a disimmunity, while FP is associated with primary resistance to immunotherapy [28,55,59].

No specific therapies exist for HPD or FP patients. Chemotherapy, TKI, clinical trial or best supportive care are currently the only possible treatment options. Re-biopsy to search for targetable alterations as resistance mechanisms or histology transformation should be considered for treatment decision and also for implementing biological knowledge about this phenomenon [60,61,62].

We are going to describe the major recent studies on the systemic strategies for ES-PD patients treated with ICI–CT.

## 6. Treatment Option Strategies for Early Systemic Progression

In this section, we reported different strategies for overcoming resistance that leads to ES-PD and IS-PD including (1) strategies with immunotherapy, (2) strategies beyond immunotherapy, and (3) innovative trials with different multiple approaches.

### 6.1. Strategies with Immunotherapy

The therapeutic strategies with the use of immunotherapy beyond progression included the use of (1) second-generation immunotherapeutic agents, (2) the combination of immunotherapy with antiangiogenic agents, and (3) the combination of immunotherapeutic agents.

#### 6.1.1. Second-Generation Immunotherapeutic Agents

##### IL-2 Agonist

The immune-stimulating activity of the cytokine IL-2 is well-known and NKTR-214 (Bempegaldesleukin) is an IL-2 agonist targeting CD122 receptor, which is the IL2 receptor β-subunit, found on the surface of CD8+ T cells and natural killer cells [63,64]. NKTR-214 increases the activation and the PD-L expression of these immune cells [65].

Several studies are currently ongoing in aNSCLC patients, investigating the role of this drug (PROPEL and PIVOT-02 studies).

The PROPEL study is an ongoing phase I/II multicenter study investigating the safety and efficacy of NKTR-214 combined with pembrolizumab in different advanced solid malignancies including aNSCLC. The dose optimization cohort regards the first- and second line, while the dose-expansion cohort includes first-line aNSCLC patients regardless of PD-L1 expression [NCT03138889].

The PIVOT-02 is a four-part study that evaluates the combination of NKTR-214 with nivolumab (part 1), with or without different chemotherapeutic agents (part 2) and with nivolumab and ipilimumab (parts 3 and 4). A separate cohort of part 2 will evaluate NKTR-214 with nivolumab in aNSCLC patients treated with first-line ICI–CT (NCT02983045) [66] (Table 4).

##### ICOS Receptor Agonist/Antagonist

The inducible T cell co-stimulator (ICOS; CD278) belongs to the CD28/CTLA immunoglobulin superfamily and is a positive regulator of T cells [67]. It is weakly expressed on resting Th17 and Treg cells but highly expressed on CD4+ and CD8+ T cells [68,69,70].

The ICOS agonists/antagonists are studied to overcome resistance to ICIs in monotherapy or combination with other drugs [71].

An ongoing randomized phase II trial is assessing ICOS-agonist antibody (GSK3359609) plus docetaxel versus docetaxel in aNSCLC patients progressing on ICI and CT in the same line or as separate lines of therapy (NCT03739710).

Another phase I/II trial is recruiting untreated and pre-treated aNSCLC patients for an anti-ICOS (KY1044) in combination with atezolizumab (NCT03829501).

#### 6.1.2. Antiangiogenic Agents

Receptor tyrosine kinases are known to mediate immunosuppressive mechanisms in TME, and their activation is a potential resistance mechanism to immunotherapy, while their inhibition induces an increase of the anti-tumor immune response [72,73]. This rationale suggests that combining immunotherapy with TKI may result in a re-sensitization to immunotherapy [74,75,76]. This is an opportunity for beyond progression strategies or for bypassing resistance to immunotherapeutic agents in poorly immunogenic disease sites (e.g., liver metastases) [77,78].

##### Lenvatinib

Lenvatinib is a multiple TKI that selectively inhibits VEGFR1-3, FGFR1-4, PDGFRα, c-KIT, and RET [79]. Lenvatinib has an immune-modulating effect on TME, including the decrease of tumors-associated macrophages (TAMs) and activation of cytotoxic T cells [80]. Therefore, Lenvatinib has shown to be an effective partner in combination with PD-1/PD-L1 inhibitors in different tumors types in both clinical and preclinical studies [73,80].

The ongoing phase III randomized LEAP-008 trial has the aim of assessing the efficacy and safety of pembrolizumab combined with lenvatinib versus docetaxel in non-squamous aNSCLC patients who failed after platinum-doublet CT and immunotherapy (NCT03976375).

A phase Ib/II trial is ongoing on lenvatinib plus pembrolizumab in different tumors and in its phase II part, the cohort including aNSCLC patients experienced promising efficacy results (ORR at 24 weeks of 33%) (NCT02501096) [81].

##### Sitravatinib

Sitravatinib is a spectrum-selective TKI including MET, TAM family (Tyro3, AXL, MERTK), VEGFR, PDGFR, KIT and RET [82]. The SAPPHIRE study is an ongoing phase III trial on the combination of sitravatinib and nivolumab versus docetaxel in aNSCLC patients pretreated with ICI and platinum-based CT in combination or sequence (NCT03906071) [83]. Twenty-one (84%) out of 25 patients experienced tumors reductions and 7 (28%) patients a partial response [83].

##### Cabozantinib

Cabozantinib is a potent inhibitor of multiple receptor tyrosine kinases, including MET, VEGFR, AXL and RET [84]. Preclinical studies showed that cabozantinib promotes an immune-permissive TME through the inhibition of immune-suppressive cells and tumors neovascularization [74,75,85].

It has been observed to overcome immunotherapy resistance by the resensitization to ICIs in several clinical studies and different types of tumors [86,87]. A phase Ib/II trial is ongoing on cabozantinib and atezolizumab as monotherapy or in combination as first- or further-lines in patients with multiple tumors types, including aNSCLC (NCT03170960). Finally, CONTACT-01, the ongoing phase III trial will evaluate the association of cabozantinib+atezo versus docetaxel in NSCLC pretreated with CHT and ICI (NCT04471428).

#### 6.1.3. Combination of Immunotherapeutic Agents

##### Nivolumab plus Ipilimumab

Nivolumab, an anti–PD-1 antibody, and ipilimumab, an anti–CTLA-4 antibody, modulate effector T cell activation, proliferation, and function with distinct but complementary mechanisms [88]. Their combination has proved to be effective in aNSCLC, melanoma and renal cell carcinoma [89,90,91,92].

An ongoing phase II trial is evaluating if the addition of ipilimumab to nivolumab after primary resistance to anti-PD1 therapy can lead to tumor reduction. The investigators will primarily enroll aNSCLC patients who have experienced progression or stable disease less than 24 weeks as best clinical response to anti-PD-1 monotherapy (primary resistance). A smaller cohort of patients with stable disease for at least 24 weeks, partial/complete response as the best clinical response to anti-PD-1 monotherapy, with subsequent progression (acquired resistance), will additionally be accrued (NCT03262779).

### 6.2. Strategies beyond Immunotherapy

The therapeutic strategies with the interruption of immunotherapy include the use of (1) CT in combination with antiangiogenetics, (2) CT alone and (3) the use of new TKIs.

#### 6.2.1. Antiangiogenetics plus Chemotherapy

The efficacy of the association of antiangiogenetics and CT is well-known in aNSCLC patients, such as the combination of bevacizumab with CT and immunotherapy as first-line therapy in the IMpower150 trial or the association of bevacizumab plus paclitaxel or nintedanib plus docetaxel in pretreated non-squamous NSCLC patients [93,94,95]. Moreover, antiangiogenetics have an immune effect similarly to TKIs, so their combination with immunotherapy has been investigated a different type of tumors [96,97].

The VARGADO trial is an ongoing observational prospective study that evaluates the combination of docetaxel plus nintedanib as second line after first-line CT or ICI-CT or as third line after first-line CT and second-line ICI [NCT02392455]. Grohè et al. [98] reported the results of the clinical benefit of nintedanib plus docetaxel after ICI therapy progression, according to PFS, ORR, and DCR (5.5 months, 58% and 83% respectively). This result highlighted the potential clinical benefit of treatment sequencing with antiangiogenics and chemotherapy after immunotherapy [99].

#### 6.2.2. Chemotherapy

The stimulating effect of CT on the immune system is well-known including immunogenic cell death with the release of tumors antigens in the TME, inhibition of tumors neovascularization and modulation of the immunogenicity of tumor cells by enhancing antigen presentation, upregulating expression of costimulatory molecules or downregulating inhibitory checkpoint molecules [100,101,102].

Chemotherapy and immunotherapy are known to have synergistic effects and ICI may enhance CT efficacy when delivered before the cytotoxic agent in NSCLC patients [41,103].

In the KEYNOTE-024 study, which randomized aNSCLC patients with PD-L1 ≥50% into first-line CT or pembrolizumab, cross-over was permitted on disease progression and this allows to assess the combination of the PFS for the first- and second-line therapy (PFS2) between the two arms [104]. A recent analysis showed that the PFS2 for first-line pembrolizumab plus second-line CT was significantly longer than that of first-line CT plus second-line pembrolizumab [105].

This result could lead to the hypothesis that the sooner immunotherapy is given the more efficacy will be reached and CT seems to be a valid salvage therapy after immunotherapy failure.

To date, in clinical practice, patients who progress upon immunotherapy received CT including platinum-based doublet, if not previously given, or docetaxel ± ramucirumab/nintedanib, gemcitabine, and pemetrexed [106,107].

The role of CT as salvage therapy after first-line immunotherapy is currently being investigated in an ongoing trial to assess the addition of CT to immunotherapy in patients who progressed upon PD-1/PD-L1 inhibitor [NCT03083808].

#### 6.2.3. New Targeted Therapies

##### KRAS Inhibitors

The *KRAS^G12C^* mutation is found in approximately 13% of lung adenocarcinomas and several ongoing trials are assessing the safety and activity of KRAS inhibitors in KRAS*^G12C^*-mutant patients with different types of tumors. These include also aNSCLC which progressed after standard treatment including chemotherapy and immunotherapy [108].

A phase I/II study evaluating AMG510 (Sotorasib) in pretreated patients with KRAS G12C-mutated solid tumors showed a favorable safety profile and interesting antitumor activity. The phase II cohort has now shown a durable response rate of 37.1%, a disease control rate of 80.6%, and a median progression-free survival of 6.8 months (NCT03600883) [109,110]. Recently, a randomized phase III study comparing AMG510 with docetaxel in 650 NSCLC patients has been activated (NCT04303780). If the results will be confirmed, it will be the registration trial. This randomized clinical trial aims to enroll around 325 patients per arm: AMG510 with docetaxel in NSCLC.

The phase II Lung-MAP trial is currently ongoing (NCT04625647). The other two KRAS inhibitors, MRTX849 and JNJ74699157, are currently under investigation in two phase I–II trials in patients with advanced KRAS*^G12C^* mutant solid tumors (NCT03785249, NCT04006301).

##### PARP-Inhibitors

PARP-inhibitors are oral small molecule inhibitors of poly (ADP-ribose) polymerase (PARP) enzymes which have a role in cellular growth, regulation, and cell repair from DNA damage. In this way, PARP inhibitors stop cancer cells from being repaired which causes the death of tumors cells [111]. The inhibition of DNA damage repair and the subsequent cell death increase tumors antigens release enhancing the immune response, supporting the rationale of combining PARP-inhibitors and ICIs [111,112].

There are many ongoing phase II–III studies that combine a PARP-inhibitor (e.g., olaparib) with an anti-PD1/PD-L1 (e.g., pembrolizumab) as maintenance therapy after the first line in aNSCLC patients [113] (NCT03976323, NCT03775486).

### 6.3. Multiple Strategies and Innovative Trials

Different trials are assessing different anticancer therapies in aNSCLC patients pretreated with immunotherapy.

The HUDSON trial is an ongoing phase II umbrella study that enrols aNSCLC patients who progressed after a platinum-based CT and an anti-PD-1/PD-L1 therapy, as monotherapy or in combinations. Different drugs with different mechanisms of action are assessed in combination with durvalumab including olaparib, AZD9150 (JAK-STAT3 pathway-inhibitor), ceralasertib (ATR kinase inhibitor), vistusertib (mTOR inhibitor), oleclumab (anti-CD73), trastuzumab-deruxtecan (antibody–drug conjugate) and cediranib (anti-VEGFR-1-3) [NCT03334617] [114].

In the phase I/II CheckMate 79X study, aNSCLC patients who progressed on ICIs and CT (given either concurrently or sequentially) are randomized to docetaxel versus different nivolumab-containing combinations including nivolumab (plus ipilimumab) plus cabozantinib, docetaxel plus ramucirumab, docetaxel and lucitanib, which is a VEGFR-1-3 and FGFR-1-2 inhibitor [NCT04151563].

In recent years, the CAR-T cells immunotherapy, consisting in patient’s T cells genetically engineered to produce an artificial T-cell receptor, has reported great results in many malignancies, especially in hematologic ones [115]. In aNSCLC patients, several trials are ongoing evaluating the safety and activity of CAR-T cells in different treatment settings [NCT03525782, NCT02587689].

Other co-inhibitory receptors and cell surface ligands are under investigation including T cell immunoglobulin and mucin domain 3 (Tim-3), lymphocyte-activation gene 3 (LAG-3), and Carcinoembryonic Antigen-related Cell Adhesion Molecule 5 (CEACAM5).

T cell immunoglobulin and mucin domain 3 is a co-inhibitory receptor particularly expressed on exhausted CD8+ T cells and in preclinical models the co-block of PD(L)-1 and Tim-3 receptors has shown to be effective in solid tumors [116]. Furthermore, Tim-3 deregulation has been associated with the development of resistance to PD(L)-1 inhibition in NSCLC patients [117]. Many phase I/II studies are investigating the efficacy of Tim-3 antagonists in association with anti-PD(L)-1. Preliminary data of the phase I AMBER study on the combination of TSR-022 (anti-TIM-3 monoclonal antibody), and TSR-042 (anti-PD-1 inhibitor) showed promising clinical activity and good safety in aNSCLC patients progressed on anti-PD(L)-1 treatment (NCT02817633) [116,118].

Another ongoing phase I/II trial evaluates the safety and activity of MBG453 (Tim-3 inhibitor) with or without PDR001 (anti-PD-1, spartalizumab) in patients with advanced solid tumors, including aNSCLC patients, pretreated or not with an anti-PD(L)-1 therapy (NCT02608268). The phase II cohort on aNSCLC patients progressed upon anti–PD-(L)1 therapy receiving MBG453 + PDR001 showed good tolerance but limited efficacy [119].

A bispecific antibody inhibiting both Tim-3 and PD-1 (RO7121661) is currently studied in a phase I study in patients with advanced solid tumors including aNSCLC (NCT03708328).

Lymphocyte-activation gene 3 is expressed on activated CD4+ and CD8+ T cells, Treg and other immune cells. Similar to CD4, Lag-3 binds MHC class II, but with a higher affinity, with the subsequent reduction of T cell proliferation and lower pro-immune cytokine production [120]. There are many ongoing phase I/II trials evaluating the safety and the activity of LAG-3 inhibitors as monotherapy or in association with anti-PD(L)-1 in many advanced tumors, including aNSCLC pretreated with immunotherapy [NCT 02460224, NCT01968109, NCT02913313]. Furthermore, also for LAG-3, there is an anti-PD-1-LAG-3 bispecific antibody that is currently under evaluation in a phase I trial on patients with advanced solid tumors, including aNSCLC patients previously treated with PD-(L)1 inhibitor (NCT04140500).

CEACAM5 is a surface protein on tumors cells involved in cancer invasion and metastatization [121]. SAR408701 is an antibody-drug conjugate that consists of anti-CEACAM5 antibody conjugated to a cytotoxic agent maytansinoid DM4. The CARMAN-LC03 trial is an ongoing phase III trial on SAR408701 versus docetaxel in pretreated CEACAM5+ aNSCLC patients progressing after CT and ICIs [NCT04154956].

For more advanced immunotherapeutic agents (oncolytic viruses, vaccines, other cellular therapy) we suggest referring to dedicated reviews and make a constant bring up to date on dedicated software (e.g., ClinicalTrials.gov (accessed on 6 February 2021), PubMed).

## 7. Treatment Strategies for Late Systemic Progression

Long-responders to immunotherapy should be divided according to the timing of progression in those who progress after interruption of prior immunotherapy and those who progress during immunotherapy. Patients who progress after a therapeutic interval from immunotherapy in monotherapy or combination may benefit from treatment rechallenge of the interrupted therapy.

### 7.1. Rechallenge of Immunotherapy after Immunotherapy

The rechallenge of ICIs could be defined as a second course of treatment after an interval of almost 3 months. This because, regardless of the dose, the half-life of most anti-PD-(L)1 antibodies ranges between 12 and 20 days and the occupancy of PD-1 molecules on circulating T cells remains for almost 3 months [122]. The ICI rechallenge is a promising treatment approach, especially in advanced melanoma patients [123,124,125].

To date, three prospective clinical trials reported the efficacy and safety of ICIs rechallenge in aNSCLC patients. The CheckMate 153 investigated the survival benefit of a fixed-duration (1 year) vs. continuous treatment of nivolumab as second-line therapy in aNSCLC patients. In the fixed-duration arm, 47 patients progressed during the follow-up period and 39 patients (83%) were retreated with the same therapy [126]. The median duration of nivolumab retreatment was 3.8 months and disease progression on target lesions and new lesions were reported in 35% and 41% of cases, respectively [126].

In the phase II/III KEYNOTE-010 trial on pembrolizumab versus docetaxel in pretreated aNSCLC patients with PD-L1 ≥1%, after 2 years of pembrolizumab 25 (32%) patients progressed and 14 (56%) were rechallenged with a second course of pembrolizumab, reporting partial response and stable disease in 43% and 36% of patients, respectively, with a disease control rate of 79% [127,128].

In addition, in the first-line setting, KEYNOTE-024 trial reported a disease control rate of 70% in untreated patients with PD-L1 ≥50% receiving retreatment with pembrolizumab after the completion of 2 years of pembrolizumab [129].

Rechallenge in real life has been recently published in a national database analysis on 10,452 sNSCLC patients treated with nivolumab. About half of the patients received post-nivolumab therapy lines and among them, 1517 patients (about 30%) received a second course of PD-1 inhibitors, either after a treatment-free interval (resumption group, *N* = 1127), or after chemotherapy (rechallenge group, *N* = 390). The mOS was 15.0 and 18.4 months in the resumption and rechallenge group respectively and, regardless of the group, it was longer in patients initially receiving nivolumab for ≥3 months [130].

A phase II clinical trial is assessing rechallenge with pembrolizumab as second or further-line in aNSCLC patients progressing on anti-PDL1 drug. This trial consists of two treatment groups depending on when the progression disease occurred: cohort 1 consists of patients progressing during treatment or <12 weeks after stopping it, then received CT ≥4 cycles and progressed again; cohort 2 consists of patients who stopped treatment and progressed after ≥12 weeks (NCT03526887).

According to these data, in aNSCLC patients experiencing a long-term benefit from ICI, the rechallenge of immunotherapy can be considered as a therapeutic option, especially in case of a lack of valid therapeutic alternatives. However, available literature data are not sufficient to give clear recommendations and more prospective trial are needed.

### 7.2. Rechallenge of Chemotherapy after Immuno-Chemotherapy

Rechallenge with CT may be attempted if the disease has initially responded to it and is recommended in many tumors whenever there are no valid treatment alternatives [131].

Several phase II studies investigated the clinical benefit of platinum-based CT in patients previously treated with it with conflicting results [132,133]. The pooled-analysis conducted by Petrelli et al. [132] on 11 studies showed that rechallenge with platinum-based CT was associated with an interesting tumor response rate of 27% but with no survival advantage compared to conventional second-line agents.

The availability of different effective drugs and the potential cumulative platinum-related hematological (neutropenia, anemia, thrombocytopenia) and non-hematological toxicities (renal damage, ototoxicity, neurological toxicity, etc.) makes the platinum-based CT rechallenge an unusual strategy in clinical practice.

However, retreatment with platinum-based CT could be hypothetically proposed for patients treated with first-line ICI-CT who are still in treatment with immunotherapy and with a long time to progression from the last CT.

A prospective trial should be conducted to definitively address if platinum-based CT rechallenge after ICT-CT could represent an option for relapsed platinum-sensitive patients.

## 8. Conclusions

Identifying effective treatment strategies for NSCLC patients who have progressed upon single-agent ICI or ICI-based combinations is an unmet clinical need and an important issue of clinical research.

Many ongoing studies are investigating different approaches to overcome the different resistance mechanisms in both oligoprogressive and systemic progressive ICI patients, therefore enrollment in clinical trials is recommended.

New LAT methods and drug combinations could overcome resistances in oligo PD during immunotherapy.

In systemic PD, a new challenge is to estimate the type of resistance by reasoning about the timing of PD and, if possible, by performing a new biopsy (Figure 1).

## Figures and Tables

**Figure 1 cancers-13-01300-f001:**
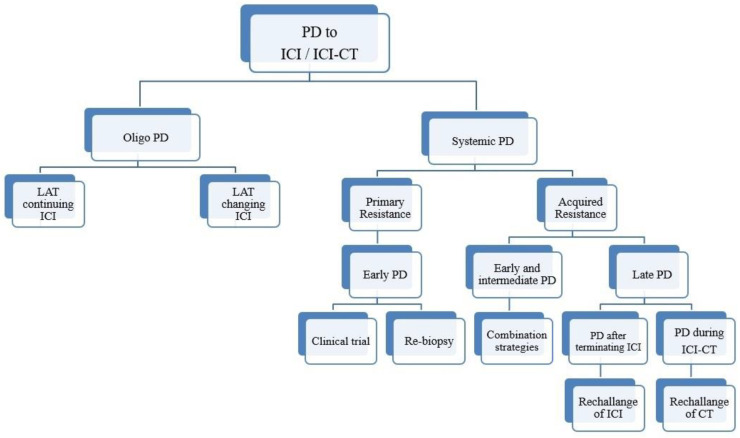
Schematic representation of an algorithm for patients with NSCLC who progressed upon IO-based therapy.

**Table 1 cancers-13-01300-t001:** Current evidence on local ablative therapy (LAT) in advanced non-small cell lung cancer (aNSCLC) patients who oligoprogressed on immune-checkpoint inhibitors (ICI).

Author (Year)	*N* Patients	ICI	ICI Line	Time to Progression (Months)	Oligoprogressive Site for LAT	Treatment Strategy	LAT	Best Disease Response	Time from Progression (Months)	PFS from ICI (Months)	OS from ICI (Months)
Case report											
Griswold (2019)	-	Pembrolizumab	I	2	Subcutaneous lesions	LAT plus continuing ICI	Surgery	Stable disease	≥12	≥15	≥15
Sotelo (2020)	-	Nivolumab	II	12	Adrenal lesion	LAT plus continuing ICI	SBRT	Partial response	≥29	≥41	≥41
Tobita (2020)	-	Nivolumab	III	17 ^a^ 6 ^b^	Bone metastasis ^a^, small intestine lesion ^b^	LAT plus continuing ICI	RT ^a^, surgery ^b^	Stable disease	6 ^a^ ≥18 ^b^	23 ^a^ ≥41	≥47
Retrospective											
Bledsoe (2016)	21	ICIs	-	median = 2.7	Bone, brain, lung	LAT plus continuing ICI (29%)	RT	LC at 6 and 12 months = 92%, 85%	median = 2.3	-	mOS = 7.2
Mersiades (2017)	10 (5 received LAT)	Pembrolizumab, nivolumab	II	median = 20.2	-	LAT plus continuing ICI (70%)	RT	LC	-	-	mOS after progression: 11.44
Gettinger (2018)	26 (15 received LAT)	ICIs (monotherapy, combination)	-	median = 10.3	Lymph node, adrenal, brain, lung	LAT plus continuing ICI (73%)	-	-	-	-	mOS = NR 2 year OS = 92%
Guisier (2019)	27	Pembrolizumab, nivolumab	I, II, III	median = 6.9	Brain, lung, bone adrenal gland	LAT plus continuing ICI (81%)	RT, surgery	-	-	13.1	.
Metro (2019)	13 (9 received LAT)	Pembrolizumab	I	-	Brain, lung, lymph node, kidney	LAT plus continuing ICI	RT	-	PPS at 6 and 12 months = 89%, 71.1%	-	-
Rheinheimer (2020)	48 (28 received LAT)	ICIs (monotherapy, combination)	≥I	Range 4–11	Brain, lung, lymph node	-	RT, surgery	-	median: 14	-	16 NR
Kroeze (2019)	108 (31% received ICI)	Nivolumab, pembrolizumab	-	-	Extracranial or cranial lesions	LAT plus continuing ICI/TKI	SBRT	LC = 12 months	At 1 y, 47% of pts continued ICI	7	Improved mOS *p* = 0.008

*N*, number; pts, patients; ICI, immune checkpoint inhibitor; RT, radiotherapy, LC, local control; PFS progression-free survival; PPS, post-progression survival; OS, overall survival; mOS, median overall survival, NR, not reached; TKI, tyrosine kinase inhibitor; SBRT, stereotactic body radiation therapy. ^a,b^ First (a) and second (b) phase treatment of nivolumab in the same patient before bone (a) and small intestine (b) progression.

**Table 2 cancers-13-01300-t002:** Ongoing trials on LAT in advanced NSCLC patients who progressed on ICI.

N of the Clinical Trial	Phase	Type	Arm(S)	Estimated Enrolment (*N* Participants)	Patients	Treatment	Primary Endpoint	Main Secondary Endpoints
NCT03158883	Early I	Interventional, non-randomized, single center	Two groups	26	Non-responders: patients who progress at first response assessment Progressors: patients who initially experience response or stable disease and subsequently progress	Avelumab 10 mg/kg IV infusion q2w + SBRT 50 Gy/5 fr	ORR	OS, PFS DCR DSD, DOR irRC
NCT04549428	II	Interventional, non-randomized, multicenter	Single-armed	20	Oligoprogressive: ≤4 PD lesions, ≤3 organs, ≤3 lesions per organ, except bone lesions	Atezolizumab 1200 mg, IV infusion every 3 weeks + Palliative RT 8 Gy/1 fr, concomitant to the 2nd dose of atezolizumab	ORR	OS, PFS
NCT04405401	II	Interventional, randomized, single center	Two groups	68	Oligoprogressive: ≤5 PD extracranial lesions, ≤5 cm and involving ≤3 organs. (PD at the primary tumors counted within the 5 lesions. Each lymph node metastasis is counted as one site of metastasis)	Experimental arm: definitive SBRT to PD lesions + current systemic therapy versus Standard of care: next systemic therapy line, BSC or continuing current systemic therapy	OS, PFS	Local control Time to next systemic therapy
NCT04485026	II	Interventional, randomized, single center	Two groups	70	Oligoprogressive: ≤4 PD lesions (PD of the primary tumors and/or regional lymph nodes counted as one lesion)	Experimental arm: hypofractionated local RT (>2 Gy per fr) to all PD lesions versus 2nd line systemic therapy	OS	PFS; TTP Time to 2nd line of systemic therapy or palliative care
NCT03693014	II	Interventional, non-randomized, single center	Single-armed	60	Oligoprogressive: ≤5 lesions either new or increase in ≥25% in the diameter of a known lesion	SBRT 27 Gy/3 fr to ≤3 PD lesions, while continuing ICI	ORR	-
NCT03406468	II	Interventional, non-randomized, single center	Single-armed	40	Patients who initially experienced CR, PR or SD under ICI monotherapy or ICI–CT combination and then PD	RT in different doses to one lesion, continuing ICI monotherapy or ICI–CT		
NCT04492969	Prospective	Observational non-randomized, single center	Single-armed	320	Oligoprogressive: ≤3 PD lesions in ≤2 organs	RT to ≥1 of PD lesions	Oligo-progression disease rate	ORR, OS

*N*, number; PD, progressive disease; IV, intravenous; RT, radiotherapy; Gy, Gray; Fr, fraction(s); SBRT, stereotactic body radiation therapy; BSC, best supportive care; ICI, immune checkpoint inhibitor; CT, chemotherapy; ORR, overall response rate; OS, overall survival; PFS, progression-free survival; DCR, disease control rate; DSD, duration of stable disease; DOR, duration of overall response; irRC, immune-related response criteria; TTP, time to progression.

**Table 3 cancers-13-01300-t003:** Definition of different types of resistance to immunotherapy.

**Primary resistance**	A clinical scenario where cancer does not respond to an immunotherapy strategy. The mechanistic basis of lack of response to immunotherapy may include adaptive immune resistance or a defect in antigen presentation and initiation of the immune response.
**Acquired resistance**	A clinical scenario in which cancer initially responded to immunotherapy but after a period of time it relapsed and progressed.
**Priming defective mechanism**	Cancer is not recognized by the immune system (defective priming). This could clinically manifest as primary resistance; rarer is a priming defect as the exclusive mechanism in acquired resistance because there are several active T cell clones.
**Adaptive immune resistance**	A mechanism of resistance where cancer is recognized by the immune system (correct priming) but it protects itself by adapting to the immune attack (defective development and consolidation of the immune response). Given the spatial and temporal heterogeneity of the cancer–tumor microenvironment (TME) interaction; this could clinically manifest as primary resistance, mixed responses or acquired resistance.

**Table 4 cancers-13-01300-t004:** Most relevant ongoing trials investigating different treatment strategies in NSCLC beyond first-line immunotherapy.

N of the Clinical Trial	Phase	Type	Drug	Arm(S)	*N*	Patients	Treatment	Primary Endpoint	Main Secondary Endpoints
NCT02869295	I/II	Interventional, non-randomized, multicenter	NKTR-214	Single-armed	40	aNSCLC progressed after a maximum of 2 lines.	NKTR-214 dose escalation	Safety tolerability	ORR; BOR; DOR; PFS; CBR; MTR; OS; PK;
NCT03138889	I/II	Interventional, non-randomized, multicenter		Two arms	135	First- and second-line aNSCLC.	NKTR-214 0.008 mg/kg d1q3w iv OR NKTR-214 0.006 mg/kg d1qq3w iv + Pembrolizumab 200 mg d1q3w iv	Safety tolerability RP2D, ORR	Safety; Tolerability ORR; DOR; CBR; TTR; PFS; OS.
NCT02983045	I/II	Interventional, non-randomized, multicenter		Four groups	557	First- and second-line aNSCLC (progressed on anti-PD-1/L1 in combination with platinum-based chemotherapy)	NKTR-214 + nivolumab OR NKTR-214 + nivolumab + platinum-based chemotherapy OR NKTR-214 + nivolumab + ipilimumab	ORR	OS; PFS; CBR; DOR
NCT03739710	II	Interventional, randomized, multicenter	ICOS agonists	Two groups	105	Advanced NSCLC progressed after a maximum of 2 lines. Anti-PD-(L1) and/or platinum-based chemotherapy (combination or sequence).	GSK3359609 80 mg d1q3w + Docetaxel 75 mg/m^2^ d1q3w vs. Docetaxel 75 mg/m^2^ d1q3w	OS	OS; PFS; ORR; DOR; safety; PK
NCT03976375	III	Interventional, randomized, multicenter	lenvatinib	Three groups	405	Stage IV NSCLC progressed on anti-PD-(L1) and a platinum-based chemotherapy (combination or sequence)	Lenvatinib 20/24 mg once a day po + Pembrolizumab 200 mg d1q3w iv OR Lenvatinib 20/24 mg once a day po vs. Docetaxel 75 mg/m^2^ d1q3w iv	OS; PFS	ORR; DOR; QoL
NCT03906071	III	Interventional, randomized, multicenter	sitravatinib	Two groups	532	Advanced non-squamous NSCLC progressed on an anti-PD-(L1) and a platinum-based chemotherapy (combination or sequence)	Nivolumab 240 mg d1q2w (or 480 mg d1q4w) iv + Sitravatinib 120 mg once a day po vs. Docetaxel 75 mg/m^2^ d1q3w iv	OS	ORR; PFS; safety
NCT03170960	I/II	Observational requential assignment, multicenter	cabozantinib	Three groups	1732	Stage IV non-squamous NSCLC progressed on or after ICI Stage IV non-squamous NSCLC PD-L1-pos in first line Stage IV non-squamous NSCLC EGFR-pos progressed on or after TKI	Atezolizumab 1200 mg d1q3w iv + Cabozantinib 20-60 mg once a day po	MTD; ORR	Safety
NCT02392455	Prospective, non-interventional	Observational, cohort, multicenter	docetaxel plus nintedanib	Single-armed	700	Second-line non-squamous aNSCLC	Docetaxel 75 mg/m^2^ d1q3w iv + Nintedanib 200 mg bid d2-21q3w po	1 year survival rate	1-year survival rate and PFS of patients with first line PD within 9 months; mOS; PFS, DCR; safety
NCT02817633	I	Interventional, non-randomized, multicenter	TSR-022 (anti-TIM-3) TSR-042 (anti-PD-1) TSR-033 (anti-LAG-3)	Thirteen groups	369	Non-squamous aNSCLC	TSR-022 OR TSR-022 + nivolumab OR TSR-022 + TSR-042 OR TSR-022 + TSR-042 + TSR-033 OR TSR-022 + TSR-042 OR TSR-022 + TSR-042 + Docetaxel	DLT, SAEs, TEAEs, irAEs, ORR	ORR, DOR, PFS, OS, PK, anti-TSR-022, anti-TSR-042 anti-TSR-033
NCT02608268	I - I b/II	Interventional, non-randomized, multicenter	MBG453 (anti-TIM-3) PDR001 (anti-PD-1)	Six groups	252	aNSCLC	MBG453 OR MBG453 + PDR001 OR MBG453 + decitabine	Safety, tolerability, ORR, DLT	BOR, OS, DOR, PFS, ORR, PK, expression of PDL-1, PDp
NCT03708328	I	Interventional, non-randomized, multicenter	RO7121661 (anti-PD-1 and anti-TIM-3)	Single arm	280	aNSCLC in first-line ICI-naive or in second/third-line (PD-L1 positive). SCLC	RO7121661	AEs, DLT, ORR, DCR, DOR, PFS.	PK, anti-drug antibodies, ORR, PDp
NCT04154956	III	Interventional, randomized, multicenter	SAR408701 (anti-CEACAM5 plus mayatasinoid DM4)	Two arms	554	Stage IV non-squamous NSCLC progressed on anti-PD-(L1) and platinum-based chemotherapy, with CEACAM5 expression	SAR408701 100 mg/m^2^ d1q2w iv vs. Docetaxel 75 mg/m^2^ d1q3w iv	PFS, OS	ORR, QoL, DOR, TEAEs, SAEs

*N*, number; IV, intravenous; RT, radiotherapy; ICI, immune checkpoint inhibitor; Aes, adverse events; ORR, overall response rate; BOR, best overall survival; DOR, duration of overall response; DLT, dose limiting toxicities; PFS, progression-free survival; OS, overall survival; CBR, clinical benefit rate; MTR, median time to response; DCR, disease control rate; TTP, time to progression; QoL, quality of life; HRQOL, health-related quality of life; ieAEs, immune-related adverse events; PK, pharmacokinetics; RP2D, recommended phase 2 dose; TEAEs, treatment-emergent adverse events; SAEs, serious adverse events; TTR, time to response; PDp, pharmacodynamic parameters.

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
