# Peer review of "Beyond First-Line Immunotherapy: Potential Therapeutic Strategies Based on Different Pattern Progressions: Oligo and Systemic Progression"

_cancers, 2021, doi:10.3390/cancers13061300_

Round 1

Reviewer 1 Report

Prelaj and co-workers presented a very interesting review entitled "Beyond first-line immunotherapy: potential therapeutic strategies based on different pattern progressions: oligo and systemic progression."

This manuscript presents and summarizes very interesting data on the therapeutic options after ICI in advanced NSCLC.

In my opinion, this manuscript is suitable for publication in Cancers Journal in the current form.

Author Response

Dear Reviewer,

please find attached the doc with my reply

Reviewer 2 Report

The authors aim to review current and potential treatment strategies in advanced NSCLC following immunotherapy failure. They divide the discussion into oligoprogression and systemic progression. This topic is important and the material they cover is very useful. While I think there is great potential for this review, I do not believe it is ready for publication.

The grammar and spelling is extremely problematic to the point it impacts the accuracy of the discussion. There are careless mistakes that also alter the findings they are reporting on. I have highlighted a few below but the authors need to carefully edit this paper for additional problems.

  • Lines 100-102 The authors state that RT is the most promising LAT because of potential abscopal effects, however the references given are for Renal cell carcinoma with RT given to the LN along with immunotherapy and Demaria’s review with mostly pre-clinical evidence. Either the statement needs to be clarified or more references need to be provided.
  • Current immunotherapies are less effective in the liver, the authors needed to address this when discussing liver mets.
  • Line 107- reword
  • Lines 110-111 – reword
  • The words “to’ and ‘on’ are misused, for example line 135 “progressing to ICIs” should be “progressing on ICIs” – this occurs in multiple places and changes the meaning of the sentence completely.
  • lines 164-168 and lines 213 -217 say the exact same thing
  • line 274 “in this chapter”??
  • Missing Greek letters in various parts
  • The words natural killer cells should not be capitalized
  • Line 287 did the authors mean to say “PD-1 expression of these immune cells”?
  • Please indicate when referring to a study not performed in humans… e.g in pre-clinical models they found that…
  • The negative effects of systemic chemotherapy need to be briefly addressed.

Author Response

Dear Reviewer,

please find attached a letter with author's reply

Reviewer 3 Report

This is a well written paper highlighting the current treatment strategies for oligo an systemic progression in  advanced non-small cell lung cancer. However it will be helpful to strengthen  by  identifying a range of challenges and opportunities facing these studies and present the summary of those challenges and opportunities in the conclusion

Author Response

Dear rEVIEWER,

Please find attached a doc with the reply

KIND REGARDS

Round 2

Reviewer 2 Report

The authors need to make additional changes, they continue to use the phrase "progressed to" when they mean "progressed on". This is critical to the meaning of the sentences. If someone progresses to something they move on to it in the future. If someone progresses on something they move on from it and on to something else. For example please see your reference "Phase II Umbrella Study of Novel Anti-cancer Agents in Patients With NSCLC Who Progressed on an Anti-PD-1/PD-L1 Containing Therapy (HUDSON)"

Please highlight changes.

Author Response

Dear Reviewer thank You again for the revisions

Please find attached our response

Kin Regards

Dr. Prelaj
